# A Graph-Attention-Based Method for Single-Resident Daily Activity Recognition in Smart Homes

**DOI:** 10.3390/s23031626

**Published:** 2023-02-02

**Authors:** Jiancong Ye, Hongjie Jiang, Junpei Zhong

**Affiliations:** 1Shien-Ming Wu School of Intelligent Engineering, South China University of Technology, Guangzhou 511442, China; 2Department of Rehabilitation Sciences, The Hong Kong Polytechnic University, Hong Kong

**Keywords:** human activity recognition, smart home, embedding, graph attention network, deep learning

## Abstract

In ambient-assisted living facilitated by smart home systems, the recognition of daily human activities is of great importance. It aims to infer the household’s daily activities from the triggered sensor observation sequences with varying time intervals among successive readouts. This paper introduces a novel deep learning framework based on embedding technology and graph attention networks, namely the time-oriented and location-oriented graph attention (TLGAT) networks. The embedding technology converts sensor observations into corresponding feature vectors. Afterward, TLGAT provides a sensor observation sequence as a fully connected graph to the model’s temporal correlation as well as the sensor’s location correlation among sensor observations and facilitates the feature representation of each sensor observation through receiving other sensor observations and weighting operations. The experiments were conducted on two public datasets, based on the diverse setups of sensor event sequence length. The experimental results revealed that the proposed method achieved favorable performance under diverse setups.

## 1. Introduction

With the development of Internet of Things (IoT) technology and the progress of deep learning algorithms, there is currently widespread interest in smart home systems using sensors for improving the daily lives of humans. For example, detecting dementia-related abnormal behaviours [1,2], analysing context [3], controlling indoor equipment automatically [4], and helping the disabled or patients to live independently better [5,6]. Human activity recognition (HAR) has been one of the research hotspots since it plays a significant role in ambient-assisted living (AAL). HAR is the process of utilizing sensors to understand human motion processes or behavioural logic, so as to infer human daily activities. If the occupant’s daily activities can be identified with smart home systems, it can satisfy the demands for smart home systems to provide rational and automated services to residents.

In recent years, several studies have focused on deep neural networks (DNNs), convolutional neural networks (CNNs), and recurrent neural networks (RNNs) to investigate the solutions for HAR [7,8,9]. For the purpose of performing the primary task of HAR, deep learning algorithms can be employed to build a model, either in a supervised or unsupervised learning way. They are among data-driven methods, taking advantage of the massive volume of data collected using the sensors available in smart home systems. With the development of IoT-based smart home platforms, this study focuses more on HAR based on environmental sensor data instead of video or wearable sensor data. The research description of IoT-based HAR is presented in Section 2.

In general, the existing deep learning methods pay more attention to inferring the relationship between daily activities and sensor observations. Indeed, the type and location of sensors, as well as user habits, will affect the performance of HAR tasks. Several practical user cases collect data using multi-type sensors such as the CASAS dataset [10]. For smart home systems with multi-type sensors that can be activated by residents or at a specific frequency, ambient sensors are irregularly activated, and the time interval between each two successive activated sensors is varied. Irregularity means that the correlation between the observed values of every two continuously activated sensors may be inconsistent. Thus, HAR challenges still remain because of the sparsity and irregularity of the data [11]. Additionally, in smart home systems, HAR is expected to assist with other tasks; therefore, real-time performance is important. However, in the state-of-the-art options available, it is difficult to maintain high accuracy when identifying activities with fewer sensor events, which will cause delays in the smart home system.

In the proposed method, we sought to model the time interval between two sensor activation events and the relationship between their location information, thus automatically extracting features to better understand the context of sensor signals, so as to deal with sparsity and irregularity. Furthermore, we aim to introduce more relevant information to represent sensor events and further improve the feature representation of sensor event sequences using the proposed algorithm. The contributions of this paper are summarised as follows:A novel representation method incorporates sensor location information and time-slice information in smart homes and generates features for each sensor event through an embedding process;We first apply the time-encoding method and vector operation to automatically capture the correlations between the sensor events relying on their timestamps;Two novel parallel modules based on graph attention (GAT), namely a location-oriented GAT module and a time-oriented GAT module, are proposed to capture the correlations between different sensor events based on location information and the activated time information. These modules automatically improve the feature representation of the sensor event sequences;A new end-to-end novel framework, namely the time-oriented and location-oriented graph attention (TLGAT) network, is established to address HAR issues. The experimental results reveal that such a model achieves superior performance on public datasets compared with other state-of-the-art methods.

In the rest of the paper, Section 2 gives a brief overview of the state-of-the-art research based on deep learning, Section 3 explains the preparation process of data from two public datasets and describes the basic principles of several modules that make up our proposed framework. Section 4 presents the experiment setup and shows the evaluation results. Lastly, we summarise the results, highlighting the superiority of the method, and draw brief conclusions in Section 5.

## 2. Related Work

In this section, we will summarise the related research of deep learning methods for HAR in smart homes and describe graph neural networks in more detail.

### 2.1. Human Activity Recognition

The task of human activity recognition has been studied for decades. In smart home systems, there are typically two HAR models: wearable-sensor-based HAR and ambient-sensor-based HAR [12]. Wearable-sensor-based HAR models use integrated sensors on mobile devices, clothes, or other portable devices, such as accelerometers, gyroscopes, and magnetometers. Ambient sensors are commonly installed at fixed locations to recognise activities, mainly including infrared sensors (e.g., RFID and PIR) and temperature sensors. In contrast, ambient sensors can detect multi-occupant activities in smart homes in a non-invasive manner for users, but it is harder to achieve accuracy using these sensors due to many influencing factors (e.g., people’s movement, people’s identity, etc.).

Currently, there are several kinds of methods of ambient-sensor-based HAR [13,14,15,16,17], including naive Bayes, hidden Markov models, support vector machines, conditional random fields, etc. Nevertheless, these traditional methods highly rely on handcrafted feature extraction techniques. With the inevitable development of deep learning in many fields, deep-learning-based studies on HAR are conducted to address the problems of feature extraction.

### 2.2. Deep Learning Algorithms

Deep learning is a promising field of artificial intelligence that is able to learn high-level features from a great volume of raw data to achieve satisfactory results in HAR. In smart home systems, the existing HAR methods focus on the preprocessing strategy of transforming a series of sensor events into specific feature representations, which further model the spatial and temporal dependencies to predict the daily activities of occupants.

Deep-learning-based methods comprise CNN-based and RNN-based methods; the former is useful to capture local dependency, i.e., the importance of nearby observations correlated with sensor events. In addition, its variants can capture either one-dimensional (1D) or two-dimensional (2D) features of event sequences with different receptive field sizes and steps. Singh et al. [18] exploited a 1D-CNN structure of raw data sequences for activity recognition and extracted the local dependencies of activity signals in an invariant feature scale. Bouchabou et al. introduced natural language-processing technology to automatically generate feature representations of raw sensor observations and applied fully 1D-CNN to improve performance [19]. In these works, the way of reducing the receptive field of convolution kernel layers by layer can achieve better outcomes. Although these works focused more on the initial representations of raw sequences, they ignored the spatial and temporal characteristics of sensor event sequences. In addition, some researchers proposed a strategy of converting sensor event sequences into 2D images for corresponding investigations. In a smart home with only binary sensors, Gochoo et al. transformed activity sequences into binary images or RGB images, to apply a 2D-CNN model [7,20]. Later, they used coloured images to represent sensor event sequences containing non-binary sensors, such as temperature sensors [20]. In comparison, Mohmed et al. utilised grayscale images instead of coloured images and selected the AlexNet structure in the feature extraction phase [21,22]. Their experiments showed that the image construction method and 2D-CNN structures cannot achieve better performance. Such constructed images have sparse pixel information, which makes it difficult for 2D-CNNs to extract local features.

RNN is another advantageous deep-learning approach to deal with sequential dependencies. Previously, Arifoglu et al. explored and compared the performance of many types of RNN models in activity recognition for elderly people with dementia [23]. Liciotti et al. summarised several types of long-short term memory variants and evaluated these models in the activity recognition task of a smart home scenario [24], proving that RNN-based approaches have excellent effects in HAR. Furthermore, Bouchabou used the NLP word-encoding method and LSTM to encode sensor observation [25]. They could achieve slight improvement with the help of word-encoding technology, and the dependency of RNNs on capturing dynamic time-varying sequences is still limited.

As introduced above, deep-learning-based methods have shown superiority in HAR. Generally speaking, the above-mentioned methods statically represent the characteristics observed with each sensor, but none of them take into account time-specific and location-specific characteristics of sensor events. In contrast, this study takes the activation moment and spatial location of the sensors into consideration, which automatically refreshes feature representations of sensor observations based on their data.

### 2.3. Graph Attention Network

The latest development, known as the graph attention network, is increasingly prevalent and has extensive applications in various fields [26], owing to its effectiveness and simplicity in transmitting and aggregating messages from neighbouring nodes in the graph structure. For instance, the data in the recommendation system can be represented as a graph structure to construct the user–item interaction [27]. Social networks can accordingly be denoted as topological graphs, to describe the relationships between people based on different factors (e.g., emails, transactions, etc.) [28,29]. Moreover, a sequence can be treated as a sequence graph, where each graph node can be connected with one or more nodes [30]. Subsequently, on the basis of the prior progress of graph convolutional networks (GCN) [31], GAT is able to produce node representations with an attention mechanism applied in each pair of nodes to distinguish the contribution of each node on the target node. A variant of the algorithm based on the key-value attention mechanism can be viewed as an extension and application of the GAT framework in activity recognition in a smart home setting. In our case, we formulate the links between sensor events as graphs, where nodes refer to sensor events and edges are the intensity of these connections. The details are described in Section 3.3.2. The variant takes advantage of GAT to determine the specific importance of neighbouring nodes for the central node, to improve the representation of each node.

## 3. Methodology

This section describes the proposed method in detail. The overall framework involves four steps, namely (i) the preprocessing of several open datasets in the CASAS, (ii) data representation, (iii) model architecture, and (iv) training and evaluation.

The flowchart of the proposed method is shown in Figure 1. The implementation process is described in detail in this section.

### 3.1. Dataset Description

In this study, two publicly available datasets in the CASAS project [10], namely ARUBA and MILAN, were used to evaluate the proposed approach. Each dataset is a file containing more than one million raw sensor events, which have been continuously recorded in the respective smart home testbed for several months. Table 1 summarises the types of sensors, the number of residents, the number of categories involved in daily activities, and the total raw sensor events in these two datasets [32,33].

Figure 2 illustrates samples of the ARUBA’s raw dataset, including annotations. One sensor event comprises a date, time, sensor ID, and sensor status or indication, which implies that a sensor is activated and changes in indication or status. In particular, the activity annotation will indicate the corresponding sensor events at the beginning or end of the activity.

Moreover, the opening datasets in the CASAS project provide the sensor layout in the smart home with image files.

### 3.2. Data Preprocessing

After acquiring the publicly available datasets, we preprocessed the data for validation experiments, and the preparation process can be divided into several phases.

#### 3.2.1. Activity Segmentation

Based on the labels provided by the dataset, a series of activity samples were extracted, each of which represented a sensor event sequence recorded from the beginning to the end of an activity. For example, a total of 6413 annotated activity segments can be segmented from the ARUBU dataset. However, the activity segments are composed of different numbers of sensor events. Some activity segments last longer than others, so the maximum and minimum number of events are 2341 and 20, respectively.

#### 3.2.2. Data Representation

**Index Representation.** The sensor ID, sensor status, or indication of a sensor event in the raw dataset are represented as one word. For example, in the ARUBA dataset, the motion sensor ID is represented as a word with four characters from ’M001’ to ’M031’. The complete word list forms a limited-size word list. To make such discrete data usable using a neural network, we transformed each word into an index. Specifically, an $N_{s}$-length word list, for recording the unique sensor ID, and an $N_{v}$-length word list for recording the unique sensor status or indication were generated, respectively. The index tokens would then be passed through an embedding layer for the continuous feature vectors learned in an unsupervised manner.

Moreover, the layouts of sensors vary in different smart home settings. As shown in Figure 3, for both datasets, the index representations based on the layout of sensors in different rooms were manually defined. For example, the layout indexes of the sensors located in the same room should be the same. The key idea is to introduce sensor layout information based on the minimal semantic information location information of smart homes ( e.g., kitchen or bathroom), as there is a strong correlation between daily indoor activities and the continuous activation of multiple sensors in a specific space. First, a smart home is divided into several subspaces according to the room layout image file provided by the dataset, and then an index is defined for each subspace. Finally, layout index information is added to each sensor event. This idea is also consistent with the human understanding of a housing layout.

Furthermore, in this study, we considered the fact that temporal features play a certain role in activity recognition because of the regularity of household daily activities. However, it is more accurate to state that there is a relationship between the activities within a certain time slice. In this method, the timeframe of 24 h is divided into 1440 time windows, and an index is assigned based on the time sequence. If the time recorded in the sensor event falls within a time slice, the assigned index is the position order of the time slice, which is defined as the time slice representation, avoiding the isolation of the temporal information contained within the slices. In general, daily household activities start at a certain time of the day and are continuously performed for a period of time, meaning that the characteristics of the activities in two different time windows may be related. Therefore, it is reasonable to use a time-slice representation to segment and analyse the temporal information.

**Time Encoding.** Herein, a high-dimension feature is generated to represent the time point but not the time slice. For each given fine timestamp *t*, it can be transformed into a multi-dimensional feature vector $p^{t}$ as follows [34]:(1)$$\begin{array}{c} p_{2k}^{t}=sin\left(\frac{t}{T^{2k/\epsilon }}\right) \end{array}$$(2)$$\begin{array}{c} p_{2k+1}^{t}=cos\left(\frac{t}{T^{2k/\epsilon }}\right) \end{array}$$
where $\epsilon \in N^{+}$ refers to the dimensionality of the generated time representation, and *T* is the time scale of the time-feature transformation, that is, the specific frequency. We set ϵ=32 and T=1440 (unit: minute) in all the experimental settings.

#### 3.2.3. Sliding Windows

It is known that a smart home system will generate an uncertain length of sensor event sequences whenever a user performs any activity. Therefore, a series of event streams should be split through a fixed-size, event-count-based window. The window size is denoted as *N*, which represents the event count in a window. It is assumed that the residents perform daily activities to generate a sensor event sequence $a_{i}$ and denote an event as $e_{j}$. As shown in Figure 4, the sliding-window method can transform an event sequence $a_{i}=\left\{e_{1},e_{2},\cdots ,e_{l}\right\}$ into $a_{i}=\left\{S_{1},S_{2},\cdots ,S_{l-N+1}\right\}$, where $S_{k}=\left\{e_{k},e_{k+1},\cdots ,e_{k+N-1}\right\}$.

Thus, using the window per sensor event generates a large number of window samples, which is helpful for obtaining more data. Obviously, setting a different window size *N*, the complexity and accuracy of the recognition model also have significant differences. Commonly, the length of the window refers to the time latency of the prediction using past sensor events. The smaller the window length is, the lower the model precision is, but the better the real-time performance of the model.

### 3.3. Model Architecture

This section introduces the proposed model architecture in detail with the purpose of learning an appropriate sensor observation embedding for a given sample $S_{k}$ to predict an activity label.

#### 3.3.1. Embedding Layer

In the first step, the prepared index representations of a sensor event $e_{i}^{ind}=\{si_{i}^{ind},o_{i}^{ind},$$l_{i}^{ind},ts_{i}^{ind}\}$ are passed through four embedding layers to a fixed-dimensional feature vector:(3)$$\begin{array}{c} si^{emb}=ED_{si}\left(si^{ind}\right) \end{array}$$(4)$$\begin{array}{c} o^{emb}=ED_{o}\left(o^{ind}\right) \end{array}$$(5)$$\begin{array}{c} l^{emb}=ED_{l}\left(l^{ind}\right) \end{array}$$(6)$$\begin{array}{c} ts^{emb}=ED_{ts}\left(ts^{ind}\right) \end{array}$$
where si,o,l,ts represent sensor ID, sensor observation, sensor location, and time slice, respectively. The superscript ${}^{ind}$ and ${}^{emb}$ are applied to distinguish between the indexed representation and the embedded representation.

#### 3.3.2. Graph Attention Layer

This section introduces the core graph attention in detail, which was used in the proposed model. Overall, the embedding process can be learned using a hierarchical architecture composed of four levels to model the irregular intervals and latent spaces of the sensor events in the samples.

Generally, a graph attention layer is adopted to model the relationships between nodes in an arbitrary graph [35]. Therefore, for a given graph and one of its nodes *u*, it is proposed that a time-oriented graph attention layer and a location-oriented graph attention layer can borrow information from node *u*’s neighbour nodes.

Firstly, a directed weighted graph $G_{i}=\left\{V,E_{i}\right\}$ is built. For every sample, $S_{k}$, node V represents a sensor event, and an edge $E_{i}$ describes the dependencies between the sensor events in sample $S_{k}$. It can be determined that there is a bidirectional asymmetric information exchange between all sensor event nodes, so the graph is a kind of complete graph. The aforementioned two proposed graph attention layers use the same way of building the weighted graph.

Secondly, let *u* indicate a sensor event, whose original observation is $X=\{si^{emb},o^{emb}\}$ and location information $l_{u}^{emb}$ at time *t* with time-slide representation $ts_{u}^{emb}$ and time-encoding representation $p_{u}$. The time-oriented and location-oriented production of observation embeddings $T_{u}$ and $L_{u}$ for sensor event *u*, as described in Figure 5, are determined using the following equations:(7)$$\begin{array}{c} O_{u}=GAT\left(J_{u}\right),\left[J_{u},O_{u}\right]\in \left\{\left[p_{u},T_{u}\right],\left[l_{u}^{emb},L_{u}\right]\right\} \end{array}$$

Specifically, first, the time-oriented graph attention layer is used to capture the relationship between *u* and its adjacent nodes to estimate the enhanced observation embedding $T_{u}$. There are *N* nodes in each sample $S_{k}$, which indicates that an *N*-th attention weight $\alpha_{u}=\left\{\alpha_{u,v}∥\alpha_{u,v}\in \left[0,1\right]\right\}$ will be generated. The intermediate LeakyReLU nonlinearity operations are omitted (the negative input slope is 0.2), and dropout operations occur (the dropout rate is 0.2), followed by a normalisation operation; thus, the attention weight is calculated using the following equation:(8)$$\alpha_{u,v}=\sigma \left(A{\left[p_{u}∥p_{v}\right]}^{T}\right)$$
where A is a trainable weight matrix shared to every two nodes *u* and *v*. $p_{u}$ and $p_{v}$ are multi-dimensional vectors using the time-encoding method. σ is a SoftMax function for the normalisation of $\alpha_{u,v}$. Later, the time-oriented GAT layer computes the enhanced representation for each node as follows:(9)$$X_{u}^{p}=\sigma \left(\sum_{v=1}^{N}\alpha_{u,v}X_{v}\right)$$

The process of the location-oriented graph attention layer is similar to that of the time-oriented graph attention layer, which can be described with the following equations:(10)$$\begin{array}{c} \beta_{u,v}=\sigma \left(B{\left[l_{u}^{emb}∥l_{v}^{emb}\right]}^{T}\right) \end{array}$$(11)$$\begin{array}{c} X_{u}^{l}=\sigma \left(\sum_{v=1}^{N}\beta_{u,v}X_{v}\right) \end{array}$$
where B is the trainable weight matrix in the location-oriented graph attention layer. It applies the location information to estimate the appropriate observation embeddings $L_{u}$.

Finally, high-level sensor event representations are obtained by concatenating $X_{u}^{p}$ and $X_{u}^{l}$.

### 3.4. Overall Network Architecture

The overall network architecture based on GATs is presented in Figure 6, which includes several modules that are arranged in sequence. More details are provided in this section.

The proposed method applies a total of four embedding layers $[ED_{si},\;ED_{o},$
$ED_{ts},\;ED_{l}]$ in the first layer, each of which converts each index representation into a 32-dimensional vector;The original embedding representation of sensor ID and sensor observation is processed with two parallel multiple GAT layers, which automatically optimise the characteristic of sensor events considering time-wise and location-wise specificity;The outputs of two parallel multiple GAT layers are connected with the location representation and time-slice representation. Then, they are passed through a four-layer 1-D convolution layer with a kernel size of 5, followed by an average pooling layer to capture the high-level features of the sensor event sequence for the inference of daily activities;The outputs of the average pooling layer are fed to the fully connected layer to infer activity categories.

### 3.5. Model Complexity Analysis

This section introduces the computational complexity of the proposed model in detail. As mentioned in Section 3.2, the sensor event count in a window is denoted to *N*, which refers to the length of the sensor event sequence. In addition, *O* is commonly used to describe the asymptotic behaviour of the functions that define the time cost of the algorithms in the computer science field. $F_{i}$ and $F_{i}^{\prime }$ are used to represent the dimension of input and output vectors of the $i_{th}$ layer, where i=1,2,3 refers to the embedding layer, the GAT layer, and the convolutional layer, respectively. *C* refers to a known constant.

Firstly, the embedding layer converts indexes into vectors. Since the number of indexes is constant, the computational complexity of an embedding layer is $O(NF_{1}^{\prime })$. Afterwards, the GAT changes the feature dimension of input vectors through matrix operation and computes the coefficient between every two sensor events, so its computational complexity is $O(NF_{2}F_{2}^{\prime }+N^{2}F_{2}^{\prime })$. In addition, the computational complexity of a 1D convolutional layer is quadratic with kernel size (*K*), the numbers of channels ($F_{3}$ and $F_{3}^{\prime }$) as well as the length of input vector (*N*), and its computational complexity is actually $O(NF_{3}F_{3}^{\prime }K^{2})$. Therefore, the total computational complexity of the proposed model is described as follows:(12)$$\alpha_{1}\times \left(NF_{1}^{\prime }\right)+\alpha_{2}\times O\left(NF_{2}F_{2}^{\prime }+N^{2}F_{2}^{\prime }\right)+\alpha_{3}\times O\left(NF_{3}F_{3}^{\prime }K^{2}\right)$$
where α refers to the coefficient of each formula.

## 4. Experiments

### 4.1. Experimental Setups

First, as previously described, the datasets were segmented using the sliding-window method with different window sizes. In general, the cost of the proposed methods and other existing methods are mostly related to the sequence length. Therefore, the smaller the sequence length, the faster an activity can be recognised. To verify the performance of such models under small sequences, the window sizes were set to 20, 25, 30, and 35 to verify the model performance in various latencies.

Next, to evaluate the proposed method, the data samples of each category were split into two parts, i.e., 70% for training and 30% for testing. Random-shuffle processing ensured that each model could be trained to achieve better generalisation performance.

Then, the proposed TLGAT method was compared with the state-of-the-art models for human daily activity recognition in publicly available smart home systems, including embedding-based FCNs (E-FCNs) [19], ImageDCNN [7,20], and ELMoBiLSTM [25]. As far as our method is concerned, the hidden dimension sizes of the embedding layer and the convolutional layer were empirically set to 32 and 128. Other hyperparameter settings are presented in Table 2. In particular, the Adam optimiser was used to train all the models in a specific number of iterations with an initial learning rate (LR) of 0.01, which was delayed at a specific rate during training. Each experiment was performed 30 times, with a trust gap of 99% using the statistical calculation based on three-sigma (3σ) limits.

Eventually, all the experiments were carried out on a server, with an 11th Gen Inter(R) Core(TM) i7-11700K @3.60GHz, 32GB RAM, and an NVIDIA GeForce RTX 3070(8GB). Torch frameworks were used for the algorithm’s implementation.

### 4.2. Training and Evaluation

The overall loss for training was computed using the cross-entropy function, which is a suitable measurement option for the difference between the predicted and true probability distributions.

In this task, the sample share of each category was imbalanced. The performance of the models was measured using the $F_{1}$-score (mean).
(13)$$\begin{array}{c} Precision=\frac{TP}{TP+FP} \end{array}$$
(14)$$\begin{array}{c} Recall=\frac{TP}{TP+FN} \end{array}$$
(15)$$\begin{array}{c} F_{1}=2\times \frac{Precision\times Recall}{Precision+Recall} \end{array}$$
where TP, TN, and FN refer to true positives, true negatives, and false positives, respectively.

### 4.3. Comparison with State-of-the-Art Models

To show the satisfying performance of the proposed model, we compared the $F_{1}$-score of this model and other state-of-the-art models with the Aruba dataset. Table 3 and Table 4 show the overall results of the overall performance under different sensor event window sizes. In total, the proposed model outperformed the traditional state-of-the-art models in HAR under the delay of a different number of sensor events. The $F_{1}$-score increased as the sensor event window size increased. When the length of the sensor event sequences was no more than 35, longer sequences could provide more information for these models to infer the activities. Furthermore, it seems that TLGAT can automatically generate more relevant features than other state-of-the-art methods on small sequences and can further provide a better recognition performance.

Furthermore, the $F_{1}$-score values of all activities were determined except for “Enter Home” and “Leave Home” of the ARUBA dataset as well as “Bed to Toilet” and “Eve Meds” of the MILAN dataset (for which there were a low number of samples for testing); the results are shown in Table 5 and Table 6. It can be observed that this model performed well in the classification of each activity. In particular, it seems that it was more difficult to infer some activities using the small sensor event sequences with lengths of 20 and 25 because these activities were more complex than others. If more than 30 sensor events were used, the TLGAT could better represent the sensor event sequence and improve the performance of inferring complex activities.

### 4.4. Ablation Experiments

In this work, the proposed model consists of several modules based on GAT and CNN. To validate the effect and importance of the proposed GAT-based modules, i.e., the time-oriented and location-oriented GAT modules (measured with the $F_{1}$-score on the test set), we separately trained four kinds of models, namely TLGAT, LGAT, TGAT, and baseline, in the ARUBA and MILAN datasets. Among them, the first model is our final model architecture, the second and third models are the versions of the first one after removing the time-oriented GAT or the location-oriented GAT module, respectively, and the last model removes both of these two modules. Once the architectures of the above models were identified, ablation studies were performed on the datasets.

Each trial was performed using the parameters defined in Section 4.1, so as to obtain the model with optimal performance through training. Table 7 and Table 8 demonstrate the experimental results of the four models under different sensor event window sizes in the ARUBA and MILAN datasets separately. Obviously, compared with the baseline CNN model, the improvement in the $F_{1}$-score of the models using the time-oriented or location-oriented GAT module could slightly provide positive improvement in recognizing activities to some extent. These two modules can learn the temporal and spatial correlations between different sensor events from the sensor sequence, so as to improve the expression of the sensor sequence. Moreover, merging these two modules with a feature concatenation strategy can effectively leverage the superiority of both modules. Therefore, TLGAT was found to have favourable performance in HAR.

## 5. Conclusions and Discussion

This paper introduces a novel GAT-based model, namely TLGAT, for HAR in IoT-based smart home systems. TLGAT is able to learn the time-oriented and location-oriented dependencies between sensor events and leverages a feature concatenation strategy, thus significantly improving the recognition precision of HAR tasks. The ability of the graph attention structure enables TLGAT to naturally capture the correlations between different sensor events about irregularly activated time intervals and the layout of the sensors. We validated the effectiveness of the proposed architecture on two public datasets, and extensive experiments showed that it consistently outperformed three state-of-the-art models. In addition, through ablation studies, it was found that this framework applied rational strategies to integrate the proposed GAT modules as well as achieved remarkable improvement. Therefore, GAT can provide a better understanding of HAR tasks.

In addition, there are still several challenges in multi-resident HAR tasks because of data complexity, redundancy, and heterogeneity. We plan to explore the GAT-based methods of understanding data and address these challenges. Moreover, we aim to evaluate other methods such as fuzzy Windows methods [36], to explore a novel framework to improve HAR tasks in smart homes.

## Figures and Tables

**Figure 1 sensors-23-01626-f001:**
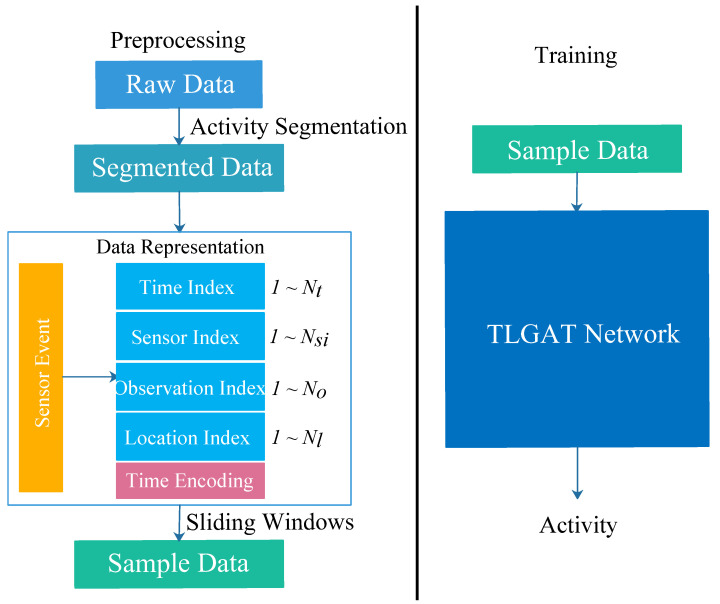
Flowchart of the proposed method.

**Figure 2 sensors-23-01626-f002:**
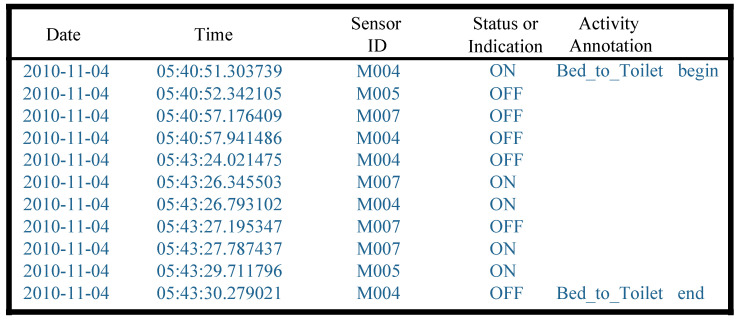
A sample of ARUBA’s raw dataset.

**Figure 3 sensors-23-01626-f003:**
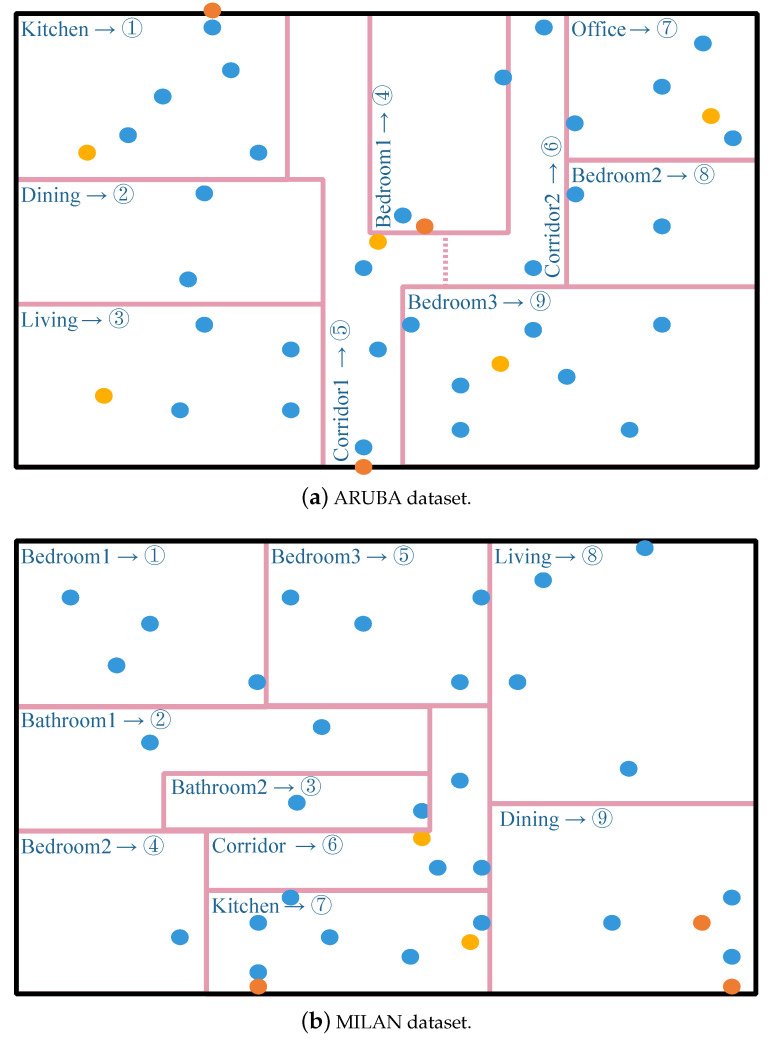
The sensor layout and location definition; blue circles are the motion sensors; yellow circles are the temperature sensors; and orange circles are the door sensors. Arrows indicate that the location indexes of the sensor in the specified area are set to the corresponding numbers.

**Figure 4 sensors-23-01626-f004:**
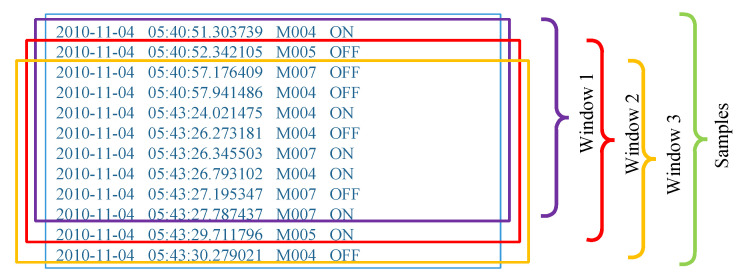
Sliding-window method.

**Figure 5 sensors-23-01626-f005:**
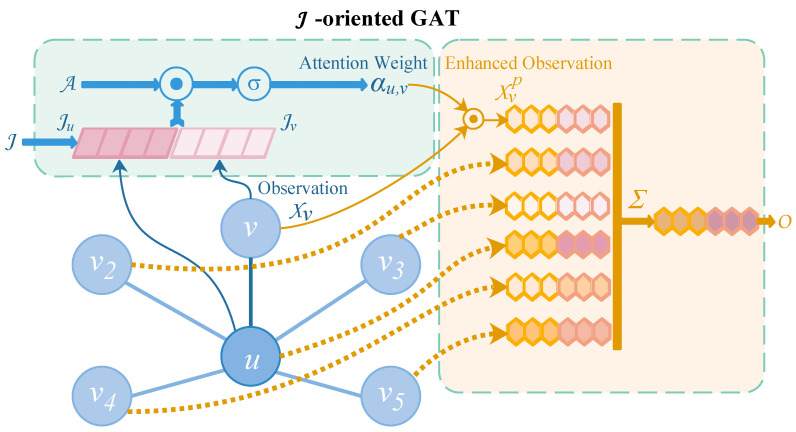
The architecture of *J*-oriented graph attention layer. In the time-oriented graph attention layer, it is assumed that $\left[J,J_{u},J_{v},O\right]$ is the time-encoding feature vector $\left[p,p_{u},p_{v},T\right]$, while in the location-oriented graph attention layer, we assume $\left[J,J_{u},J_{v},O\right]$ to be the location-embedding representation $\left[l^{emb},l_{u}^{emb},l_{v}^{emb},L\right]$.

**Figure 6 sensors-23-01626-f006:**
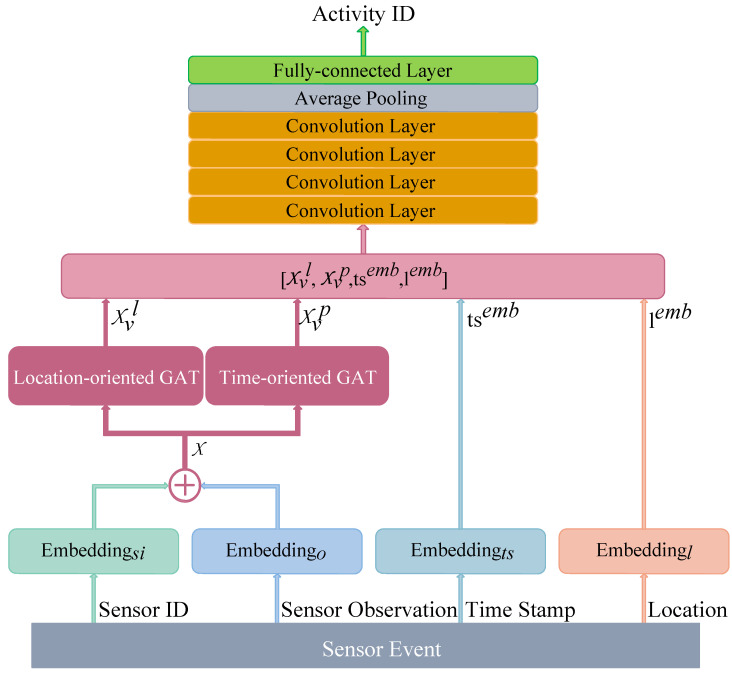
The overall network architecture of the proposed TLGAT.

**Table 1 sensors-23-01626-t001:** Dataset description (D = door switch sensor, M = motion sensor, T = temperature sensor).

Dataset	The Kind of Sensors	The Number of Residents	The Number of Raw Sensor Events	The Number of Activity Categories
ARUBA	D/M/T	1	1,719,558	11
MILAN	D/M/T	1+pet	433,665	15

**Table 2 sensors-23-01626-t002:** Training hyperparameter settings.

Training Hyperparameters	Values
Batch size	256
Dropout rate	0.2
Channels of embedding layers	32
Channels of convolutional layers	128
Learning rate	$1\times 10^{-2}$
Iterations	ARUBA: 90; MILAN: 40
Decayed epochs of LR	ARUBA: 24/48/72; MILAN: 20
Decayed rate of LR	5
Number of replications	30

**Table 3 sensors-23-01626-t003:** The overall $F_{1}$-score(%) (mean ± 3σ) of ELMoBiLSTM, ImageDCNN, E-FCNs, and TLGAT in the ARUBA dataset.

Model	N = 20	N = 25	N = 30	N = 35
ELMoBiLSTM [25]	85.307 ± 0.805	86.867 ± 0.805	89.564 ± 2.217	90.515 ± 2.855
ImageDCNN [7,20]	84.668 ± 1.864	87.618 ± 2.605	92.453 ± 2.984	95.711 ± 1.654
E-FCNs [19]	96.987 ± 0.436	98.463 ± 0.250	99.108 ± 0.098	99.442 ± 0.128
TLGAT	**97.651 ± 0.254**	**98.942 ± 0.104**	**99.514 ± 0.062**	**99.760 ± 0.199**

**Table 4 sensors-23-01626-t004:** The overall $F_{1}$-score(%) (mean ± 3σ) of ELMoBiLSTM, ImageDCNN, E-FCNs, and TLGAT in the MILAN dataset.

Model	N = 20	N = 25	N = 30	N = 35
ELMoBiLSTM [25]	76.847 ± 1.685	85.357 ± 2.228	89.975 ± 2.156	92.476 ± 1.706
ImageDCNN [7,20]	67.382 ± 1.696	76.408 ± 2.780	84.932 ± 2.589	90.857 ± 1.726
E-FCNs [19]	92.901 ± 0.252	94.656 ± 0.212	95.621 ± 0.257	96.445 ± 0.154
TLGAT	**97.569 ± 0.134**	**98.081 ± 0.181**	**98.465 ± 0.174**	**98.736 ± 0.165**

**Table 5 sensors-23-01626-t005:** The $F_{1}$-score(%) (mean ± 3σ) of different activities of TLGAT in the ARUBA dataset.

ID	Activity	N = 20	N = 25	N = 30	N = 35
1	Eating	95.167 ± 2.266	98.138 ± 0.653	98.984 ± 0.367	99.328 ± 0.211
2	Housekeeping	94.397 ± 1.265	98.088 ± 0.855	98.613 ± 0.741	99.593 ± 0.422
3	Meal Preparation	95.384 ± 0.890	97.813 ± 1.141	98.940 ± 0.216	99.524 ± 0.320
4	Other	98.021 ± 1.247	99.143 ± 0.237	99.591 ± 0.103	99.835 ± 0.037
5	Relax	98.877 ± 0.465	99.564 ± 0.136	99.811 ± 0.074	99.925 ± 0.011
6	Respirate	100.000	99.978 ± 0.018	99.578 ± 0.034	100.000
7	Sleeping	99.611 ± 0.056	99.852 ± 0.043	99.914 ± 0.012	99.980 ± 0.021
8	Wash Dishes	84.513 ± 3.264	92.104 ± 2.157	96.745 ± 1.490	98.266 ± 1.873
9	Work	96.786 ± 0.344	98.901 ± 0.021	99.443 ± 0.187	99.780 ± 0.054

**Table 6 sensors-23-01626-t006:** The $F_{1}$-score(%) (mean ± 3σ) of different activities of TLGAT in the MILAN dataset.

ID	Activity	N = 20	N = 25	N = 30	N = 35
1	Chores	99.653 ± 0.212	99.751 ± 0.102	99.671 ± 0.048	99.888 ± 0.011
2	Desk Activity	99.768 ± 0.203	99.817 ± 0.012	99.864 ± 0.101	99.944 ± 0.024
3	Dining Rm Activity	99.687 ± 0.164	99.342 ± 0.187	99.795 ± 0.105	100.000
4	Guest Bathroom	99.909 ± 0.158	99.594 ± 0.248	99.805 ± 0.124	99.827 ± 0.233
5	Kitchen Activity	97.999 ± 0.129	97.994 ± 0.153	98.403 ± 0.620	98.535 ± 0.311
6	Leave Home	99.900 ± 0.089	99.730 ± 0.180	100.000	99.893 ± 0.071
7	Master Bathroom	98.098 ± 0.375	97.681 ± 0.947	97.708 ± 1.185	97.794 ± 0.674
8	Master Bedroom	96.945 ± 0.763	96.432 ± 0.125	96.822 ± 0.417	97.113 ± 0.312
9	Meditate	100.000	100.000	100.000	100.000
10	Morning Meds	94.480 ± 4.162	94.881 ± 2.964.	97.992 ± 1.765	95.928 ± 1.178
11	Other	98.277 ± 0.111	98.140 ± 0.223	98.493 ± 0.189	98.641 ± 0.126
12	Read	98.702 ± 0.219	98.576 ± 0.367	99.018 ± 0.169	99.162 ± 0.155
13	Sleep	99.514 ± 0.110	99.286 ± 0.087	99.573 ± 0.061	99.630 ± 0.127
14	Watch TV	98.806 ± 0.188	98.431 ± 0.217	98.590 ± 0.266	98.785 ± 0.235

**Table 7 sensors-23-01626-t007:** The overall $F_{1}$-score(%) (mean ± 3σ) of TLGAT, LGAT, TGAT, and baseline in the ARUBA dataset.

Model	N = 20	N = 25	N = 30	N = 35
baseline	96.999 ± 0.154	98.694 ± 0.229	99.262 ± 0.145	99.524 ± 0.076
LGAT	97.519 ± 0.249	98.834 ± 0.199	99.469 ± 0.136	99.710 ± 0.246
TGAT	97.496 ± 0.479	**98.979 ± 0.266**	99.491 ± 0.140	99.737 ± 0.242
TLGAT	**97.651 ± 0.254**	98.942 ± 0.104	**99.514 ± 0.062**	**99.760 ± 0.199**

**Table 8 sensors-23-01626-t008:** The overall $F_{1}$-score(%) (mean ± 3σ) of TLGAT, LGAT, TGAT, and baseline in the MILAN dataset.

Model	N = 20	N = 25	N = 30	N = 35
baseline	96.381 ± 0.202	97.211 ± 0.211	97.758 ± 0.207	98.161 ± 0.184
LGAT	97.233 ± 0.125	97.826 ± 0.115	98.239 ± 0.176	98.439 ± 0.151
TGAT	97.149 ± 0.287	97.843 ± 0.152	98.404 ± 0.107	98.620 ± 0.161
TLGAT	**97.569 ± 0.134**	**98.081 ± 0.181**	**98.465 ± 0.174**	**98.736 ± 0.065**

## Data Availability

CASAS datasets are publicly available. CASAS dataset can be found at casas.wsu.edu/datasets/ (accessed on 23 December 2022).

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
