# Peer review of "A Graph-Attention-Based Method for Single-Resident Daily Activity Recognition in Smart Homes"

_sensors, 2023, doi:10.3390/s23031626_

Round 1

Reviewer 1 Report

The proposed work seems to be good but still some changes if made will surely improve the quality of the paper.

The related work can be improved by providing a comparative study stating what is observed from the literature.

The related work can be rewritten.

The metrics for evaluating the proposed work can be given.

 The clarity of the figures can be still improved.

-The introduction section needs to be rewritten with much better motivation for this work. Article needs to present background of the study, needs and significance of the study in introduction section.

 -More detailed review of the (recent) literatures is expected. Specially, it is required that the previous solutions to this problem. Please explain research gap explicitly then propose research questions. Then, the advantages (and disadvantages?) of the proposed methods should be discussed.

 -What are the criteria for choosing the current experiments?

The novelty of the method is limited. The presentation (clarity/structure) is average and the description for the method is not very easy to understand.

Check the mathematical notation especially for the proposed method.

 Add a new figure to show the general procedures of the proposed method

What is the difference between your work and the other works, and you have to cite these related works:

Augmented grasshopper optimization algorithm by differential evolution: A power scheduling application in smart homes

Multi-resident activity recognition in a smart home using RGB activity image and DCNN

The conclusion in the abstract is quite strong against other researchers' work as the experimental results don't support their claim in terms of accuracy or processing time.

Details on the implementation of the method should be included

the main contribution and originality should be explained in more detail.

the motivation needs further clarification

Reviewer 2 Report

-How can this work be extended in the near future? The future scope of the paper is included in the paper.

- Add a discussion section.

- Some latest references are missing.

- What are the major implementation challenges of this work?

-What are the challenges of the proposed work?

Reviewer 3 Report

-The novelty and major contribution of the research need properly described.

-There could be a section or text on the cost of implementing the solution proposed in the manuscript.

-Add a table in section II to present the limitations of the works cited and what the authors are contributing?

-I leave some suggestions for related work: -HOsT: Towards a Low-Cost Fog Solution via Smart Objects to Deal with the Heterogeneity of Data in a Residential Environment; -Exploiting offloading in IoT-based microfog: experiments with face recognition and fall detection;

-Section IV left the manuscript to be desired. First, it is not possible to replicate the results. Second, it does not validate the proposed solution. Third, there is no statistical analysis. Finally, there is no comparison with any work in the literature to validate the proposal?

-The authors could make the proposed solution available on Github?

-It is necessary to describe the behavior of the results and not just describe the graphics?

Round 2

Reviewer 1 Report

accept

Reviewer 3 Report

The authors responded appropriately to the comments